# Epigenetic Regulation of Obesity-Associated Type 2 Diabetes

**DOI:** 10.3390/medicina58101366

**Published:** 2022-09-28

**Authors:** Hairul Islam Mohamed Ibrahim

**Affiliations:** 1Department of Biological Science, College of Science, King Faisal University, Al-Ahsa 31982, Saudi Arabia; himohamed@kfu.edu.sa; 2Division of Microbiology and Immunology, Pondicherry Centre for Biological Science and Educational Trust, Puducherry 605004, India

**Keywords:** obesity, inflammation, T2DM, epigenetics, gut microbiota, bariatric surgery

## Abstract

Obesity is becoming more widespread, and epidemics of this condition are now considered present in all developed countries, leading to public health concerns. The dramatic increases in obesity, type 2 diabetes mellitus (T2DM), and related vascular difficulties are causing a public health crisis. Thus, it is imperative that these trends are curbed. Understanding the molecular underpinnings of these diseases is crucial to aiding in their detection or even management. Thus, understanding the mechanisms underlying the interactions between environment, lifestyle, and genetics is important for developing effective strategies for the management of obesity. The focus is on finding the vital role of epigenetic changes in the etiology of obesity. Genome and epigenome-wide approaches have revealed associations with T2DM. The epigenome indicates that there is a systematic link between genetic variants and environmental factors that put people at risk of obesity. The present review focuses on the epigenetic mechanism linked with obesity-associated T2DM. Although the utilization of epigenetic treatments has been discussed with reference to certain cancers, several challenges remain to be addressed for T2DM.

## 1. Introduction

Obesity is one of the fundamental issues associated with public health concern. It affects both genders in developed and developing countries. The WHO has estimated that more than 500 million people are obese, and, among them, 346 million (70%) have diabetes [1]. An increased food intake alongside diminishing everyday physical activity and technology dependence in modern society play major roles in the development of obesity and associated diseases [2]. Obesity-induced low-grade chronic inflammation has been linked to the pathogenesis of metabolic syndrome, which is associated with insulin resistance and T2DM [2,3,4]. Metabolic syndrome includes insulin resistance syndrome (IRS), which is associated with abdominal obesity, high blood pressure, glucose intolerance, and increased rates of morbidity and mortality [5]. IRS is reported to be linked to the development of T2DM through an increase in free fatty acid flux and a decrease in insulin sensitivity in insulin-sensitive tissues, especially visceral adipose tissue [6]. The multi-layered epigenetic network is involved in subtle dysregulation of IRS2 via IRS2 DNA methylation at CpG5, and enhanced expression of miRNA hsa-let-7e-5p in the liver has been associated with type 2 diabetes [7]. Although there has been a worldwide effort to create appropriate procedures for the prevention and treatment of obesity, it is becoming remarkably hard to change its effects on human health.

The understanding of the systematic interactions between environment, lifestyle, and genetics is essential for developing effective strategies for the management of obesity (WHO). Little is known about the link between obesity-mediated T2DM and environmental factors. Genome-wide association (GWA) studies have drastically changed their focus to the recognition of common genetic variants associated with susceptibility to T2DM [8]. Although GWA studies have determined that T2DM is becoming a prominent condition linked to genetic risk factors [9], its heritability connection is only 21% when looking at all age groups [10]. This low genetic relationship, coupled with the rapid increase in prevalence worldwide, suggests a strong role for environmental risk factors. Lai et al. [11] noticed that carbohydrate induces, whereas fat inhibits, methylation of the carnitine palmitoyltransferase-1A (CPT1A) gene. It was hypothesized that carbohydrate reduces, and fat increases, the risk of developing metabolic diseases due to the presence of differential methylation patterns. A study using a rat model revealed that T1D rats had no detectable retinopathy, while T2DM rats had significant retinopathy. A further molecular study revealed that increased Rac1 promoter DNA methylation is associated with T2DM-associated retinopathy [12]. In recent years, the focus has been improved to find the vital role of epigenetic (methylation, short RNA, long non-coding RNA and microRNA) changes in the investigative reasoning of obesity research. Genome and epigenome-wide approaches have revealed associations with T2DM. The epigenomic factors can signify crosslinks between genetic variants and environmental factors and their associations with obesity risk [13].

## 2. Epigenetic Process: Epigenators, Initiators, and Maintainers

It is suggested that some types of signals may manifest in the establishment of an epigenetic state that is unceasingly heritable: (1) an “Epigenator” is a signal that travels from the outside and activates an intracellular pathway. (2) An “Epigenetic Initiator” is a signal that reacts to the epigenator and is necessary to depict the precise position of the epigenetic chromatin environment. (3) “Epigenetic Maintainer” is a signal that maintains the chromatin environment in successive generations [14,15].

### 2.1. Epigenator

The epigenator signal consists of the chromosomal changes occurring upstream of the main event. Changes in the cell’s makeup are probably what activate the epigenetic phenotype. These consist of an environmental cue or specialty, followed by signaling pathways that prompt the initiator. Once an initiator signal has been received, it is converted to an intracellular initiator route, which results in the initiator being “activated” [14]. The protein–protein interaction or alteration-based event that triggers the epigenator signaling pathway could awaken the initiator’s latent activity. The epigenator signal will be brief, remaining in the cell just long enough to cause the epigenetic phenotype but not long enough to cause further events [14,16].

### 2.2. Epigenetic Initiator

To mediate the establishment of a local chromatin environment at a precise location, the initiator interprets the epigenator signal. The initiator will characterize the region on a chromosome in which the epigenetic chromatin configuration is to be identified after being prepared by the initiator signal. A DNA-binding protein, a non-protein-coding RNA, or any other component that can specify how the chromatin structure is put together can serve as the initiator [14]. Consequently, this signal must possess some type of sequence recognition. The Initiator is generally a signal that demands self-reinforcement and self-renewal by positive response mechanisms. One functional aspect of the initiator is that, when introduced into a cell, it can be sufficient to initiate an epigenetic phenotype. Similarly, unlike the epigenator, the Initiator could continue to work with the Maintainer rather than disseminating following its activity [17].

### 2.3. Epigenetic Maintainer

Although the Maintainer can maintain the epigenetic chromatin state, it cannot start it. This signal involves numerous mechanisms, including the nucleosome location, DNA methylation, histone modifications, and histone variations [15]. The characteristic shared by maintainers is that they lack absolute DNA sequence specificity. Therefore, they can function at any chromosomal position that an Initiator recruits them to. The role of maintainers may be to maintain the epigenetic landscapes of finally differentiated cell types or to carry an epigenetic signal across the cell cycle [14,15].

## 3. Epigenetics

Epigenetics is a vital area for developmental biology/diseases. It refers to heritable changes in gene activity/expression without any alteration to DNA sequences. Despite the fact that the genomic information from an individual is the same for all cell types, epigenetics allows for the examination of the expression of genes that are specific to a cell type in light of the cell’s unique function [14]. It is well recognized that two key epigenetic modifications—chemical methylation of DNA’s cytosine residues and histone proteins linked to DNA—strongly regulate non-genetic alternations (histone modifications). Functionally, the patterns of epigenetic changes can operate as indicators for the chromatin state, gene expression, and activity (Figure 1). An endogenous mediator of genetic vulnerability and environmental exposure in common diseases has been proposed by epigenetics through the exploration of non-DNA sequence-based data that are reproduced during cell division, such as DNA methylation [17]. Additionally, epigenetic connections to prevalent diseases and their ability to be reversed with specific medicines have been investigated. Epigenetics has garnered significant interest from both scientists and the public [18,19].

Most common human diseases are explained to a very limited degree by known individual basic hereditary variations, with 10.7% of risk profile scores explained for T2DM [20]. Even within inbred mice, methylation patterns in obese and insulin-resistant animal models were shown to differ in an extended investigation. These variations are definitely due to environmental influences rather than having a genetic basis. Therefore, in the absence of any alteration in transcription due to diabetes, an environmentally induced alteration in histone acetylation has been transmitted during mitosis to a later embryonic stage [21]. This shows that, at particular genomic locations, DNA methylation levels may be integrating and moderating hereditary and environmental factors associated with metabolic illnesses [22].

Only a few epigenetic studies of T2DM and related metabolic characteristics have been conducted so far. There are methylation differences between T2DM patients and healthy controls, according to studies of pancreatic islets (18). Obese human peripheral blood leukocytes have also shown similar alterations following human Rouxen-Y gastric bypass (RYGB) (19). Recent research has shown that chromatin changes are linked to the development of both diabetes mellitus and obesity. In the present review, we investigate the reports available on both genetic and epigenetic aspects regarding obesity and its association with T2DM (Table 1 and Figure 2).

## 4. Epigenetic Regulation of Insulin Resistance

Chronically high blood sugar levels are a symptom of T2DM, a complicated illness caused by insufficient insulin release from pancreatic cells and impaired insulin sensitivity in target organs such as the liver, skeletal muscle, and adipose tissue. Obesity is one of the key environmental and genetic variables that contribute to the development of T2DM. It is interesting to note that epigenetic modifications are presumably a result of the combination of genetic and environmental variables, leading to the development of disease [33].

Mitochondrial dysfunction reflects the metabolic status, and the genes and proteins controlling mitochondrial dynamics can be dysregulated by a high glucose concentration, leading to the overproduction of reactive oxygen species and insulin resistance [33,34]. Obese individuals show significantly increased DNA methylation in the D-loop region, which is concomitant with decreased mtDNA when compared with lean subjects. Additionally, the change in mtDNA has been firmly connected with insulin resistance, but not with impaired fasting glucose or dyslipidemia (e.g., triglycerides, cholesterol, and very low-density lipoprotein (VLDL)) (26). Zheng et al. [35] revealed higher mitochondrial D-loop DNA methylation in an obese group compared with a lean group. Interestingly, the increased methylation of the D-loop was phenocopied by insulin-resistant subjects, and DNA methylation was higher (4.6-fold) than in insulin-sensitive subjects. Consequently, the increased DNA methylation in the D-loop region was associated with insulin resistance, yet this was independent of abnormal glucose and lipid levels. The decrease in the mtDNA level in obese human subjects is also associated with insulin resistance, and this change may arise from increased D-loop methylation. These results suggest that an insulin signaling-epigenetic-genetic axis is involved in mitochondrial regulation [35]. Under hyperglycemic conditions, the epigenetic modifications of PDX-1 lead to increased DNA methylation and decreased expression of PDX-1 mRNA in β-cells, which play a role in the development of T2DM [36]. The PDX-1 encoded protein activate the transcription factor of several genes, including insulin receptor and secretion, somatostatin, glucokinase, islet amyloid polypeptide, and glucose transporter type 2. Nuclear invaded PDX-1 is involved in the pre-mature development of the pancreas and negatively regulates insulin gene expression. Mutated PDX-1 induces recurrence of insulin-dependent diabetes mellitus (IDDM), as well as mature of onset diabetes of the young type 4 (MODY4) [36].

Additionally, people with a high genetic predisposition for T2DM are portrayed to have diminished physical fitness [37]. Clearly, exercise is advantageous as it enhances insulin sensitivity and can prevent or delay the initiation of T2DM. previous research investigated genome-wide DNA methylation in human skeletal muscle from subjects with or without a family history of T2DM; in other words, they investigated whether there is a genetic predisposition for the disease using an exercise intervention study [33]. First, Differential methylation was conducted at baseline for genes involved in mitogen-activated protein kinase (MAPK), insulin, ‘Wingless/Integrated (Wnt), and calcium signaling. In addition, genes that are important for muscle functions such as (MAPK1), Myosin XVIIIB (MYO18B), Homeobox C6 (HOXC6), and AMP-activated protein kinase subunit (PRK AB1) were compared between individuals with or without a family history of Type 2 diabetes. Of the genes differentially methylated because of a family history of T2DM, 40% were validated in skeletal muscle from monozygotic twin pairs with conflicting disease statuses. Next, differentially methylated genes (134 genes) were observed after six months of exercise intervention, including Myocyte Enhancer Factor 2A (MEF2A) (exercise transcription factor), Thyroid adenoma-associated protein (THADA) (T2DM candidate gene), NADH:ubiquinone oxidoreductase subunit C2 (NDUFC2) (mitochondrial function), and Interleukin-7 (IL7) (cytokine). Subcutaneous injection of a medication known as a glucagon-like peptide-1 receptor (GLP-1R) agonist has been used to treat type 2 diabetes. Exendin-4 (Ex-4) is a GLP-1R agonist that reduces insulin resistance by directly promoting adiponectin secretion through the protein kinase A pathway [34]. Hypomethylation of the GLP-1R promoter close to the area has been observed to influence GLP-1R in grape seed proanthocyanidin extract (GSPE), which has antiobesity potential. The binding location for the SP1 transcription factor has also been investigated [35]. FOS-like 2, AP-1 transcription factor subunit (FOSL2), a potential gene linked to muscle glycogen, was discovered (MG). In the pathogenesis of T2DM, FOSL2 protein and mRNA levels were downregulated, and the DNA became hypermethylated [36].

Global DNA methylation in monozygotic twins and the connections to insulin resistance were examined by Zhao et al. [37] through a homeostasis model assessment. After adjusting for confounding factors, such as the number of established risk factors for insulin resistance, all four CpG sites examined were shown to have substantial associations with insulin resistance. The exact cause of this relationship is unknown, but scientists have hypothesized that it may be a result of genomic instability brought on by the changed methylation of the Alu sites. The elevated methylation of protein tyrosine phosphatase, a non-receptor type 1 (PTPN1) gene promoter, has been recognized as a risk factor for T2DM in the female Chinese population [38]. The loci of PTPN1 gene is on chromosome 20q13, polymorphic nucleotides PTPN1 IVS6 + G82A polymorphism displayed elevated Body mass index (BMI), percent body fat, plasma leptin, and amount of subcutaneous fat. Diet-mediated PTPN1B regulation and the development of leptin resistance has been explored using mouse models of diet-induced obesity [39]. Simar et al. [40] took things a step further and looked into global DNA methylation in several blood cell types. With this method, they discovered elevated global DNA methylation in natural killer and B cells from T2DM patients, which was closely connected to insulin resistance. Some inflammatory markers potentially play the adipocytes and insulin resistance. Monocyte chemoattractant protein-1 (MCP-1) is a monomeric chemokines protein secreted by various human adipocytes, well-known chemokine of CCR2 protein family members and is frequently involved in inflammatory stimuli such as IL-1, IL-4, or TNF-α. However, it extends beyond cell recruitment. The in vivo receptor for MCP-1 is CCR2. Importantly, prolonged exposure of adipocytes with exogenous CCR2 chemokines inactivates the lipid accumulation and PPAR-γ expression and reciprocally increased leptin secretion from mature adipocytes. These observations suggest that chemokines may have important biological effects on adipocytes and insulin resistance [41]. This characteristic emphasizes the need for tissue-specific epigenetic analyses as well as cell-type-specific epigenetic analyses, and the findings strengthen the connection between DNA methylation and immune function and metabolic diseases [42].

## 5. Epigenetics in Gestational Diabetes Mellitus (GDM)

The predominance of obesity and GDM in women is growing worldwide. GDM develops during pregnancy, most commonly in the late second trimester when the pancreatic function (impaired beta cell function) is not adequate to control the diabetogenic environment with increasing adiposity and insulin resistance [43,44]. Depending on ethnicity and scientific criteria, GDM affects 2 to 10% of all pregnancies. Environmental factors such as lifestyle, nutrition, physical idleness, and genetic risk factors explain the current GDM epidemic [44]. Differentially methylated genes, including leptin (LEP), adiponectin (ADIPOQ), ATP-binding cassette transporter ABCA1, the glucose transporters GLUT1, GLUT3, and GLUT4, and the imprinted MEST gene, have been identified in fetal tissues of babies from GDM mothers.

Bouchard L et al. [45] reported that Leptin gene (LET) DNA methylation level was increased in placenta tissue, which was highly correlated with glucose levels in women with GDM. At the same time, the DNA methylation of the adiponectin gene (ADIPOQ) was significantly decreased in fetal placenta tissue. The lower DNA methylation level of ADIPOQ was correlated with maternal hyperglycemia [46]. Boris Novakovica et al. [47] analyzed genome-scale DNA methylation and gene expression data to examine the role of methylation in glucose transporter (GLUT) expression throughout gestation. The authors revealed that DNA methylation across the promoter region of GLUT3 is involved in glucose back-flux from the fetal circulation into the placenta [47]. In addition, hypo-methylation profiles and increased expression of Retinol Binding Protein 4 (RBP4), GLUT3, Resistin, and Peroxisome Proliferator Activated Receptor Alpha (PPARα) were found in the placenta tissue of the GDM group [48]. The significant hypomethylation of the GLUT3 promoter increases its expression and also increases glucose transport during placental development, which results in sustained high glucose conditions in the placenta in women with gestational diabetes [48]. The deacetylation of histone tails with histone deacetylase 5 (HDAC5) at glucose transporters (GLUT4) in human skeletal muscle results in a condensed chromatin structure and subsequently reduces Glut 4 expression [49]. The ATP-binding cassette transporter ABCA1 gene DNA methylation levels in the placenta of women with impaired glucose tolerance are highly correlated with maternal high-density lipoprotein cholesterol (HDL-C) levels and glucose levels [50]. This study suggested that the epigenetic variations in the placenta and cord blood contribute to maternal–fetal cholesterol transfer and also potentially trigger the long-term susceptibility of the newborn to dyslipidemia and CVD [50].

Del Rosario et al. [51] identified genes with differentially methylated promoters mainly associated with MODY, T2DM, and Notch signaling in peripheral blood leukocytes between the offspring of diabetic mothers and the offspring of nondiabetic mothers. The maternally imprinted MEST gene, the non-imprinted glucocorticoid receptor NR3C1 gene, and interspersed ALU repeats showed significantly decreased methylation levels in cord blood and placenta tissue of dietetically treated GDM and insulin-dependent GDM groups compared with non-GDM groups [52]. This study suggested that intrauterine contact with GDM has lifelong effects on the epigenome of the offspring. Specifically, MEST epigenetic alternation contributes to obesity predisposition throughout life [52]. Michalczyk et al. [53] evaluated the epigenetic markers across pregnancy and the early postpartum period and measured histone H3 demethylation in the white blood cells of nondiabetic women, women with GDM who developed postpartum T2DM, women with GDM without postpartum T2DM, and women with T2DM. The demethylation of H3K27 and H3K4 was decreased in women with GDM who developed postpartum T2DM. The authors suggested that the variation in the demethylation of H3K27 and H3K4 may be a tool to predict who will develop T2DM from GDM [53].

Several genome-wide methylation studies have also confirmed that metabolic disease pathways, the inflammatory response, MAPK signaling, cell growth, death regulation, and endocytosis are epigenetically programmed through GDM exposure in the offspring of GDM mothers [44,48]. A recent study reported that GDM is associated with significant fetal cord blood (FCB) methylation changes in ATP5A1, MFAP4, PRKCH, SLC17A4, and HIF3A. This FCB methylation was more noticeable in women with insulin-dependent GDM than in women with dietetically treated GDM [44]. Thus, the underlying epigenetic mechanism increases the effect on β-cell function in the offspring of mothers with diabetes during pregnancy and increases the risk of T2DM [51].

## 6. Role of Diet in Epigenetics of Obesity and T2DM

There are now three main goals in relation to epigenetic research in the disciplines of obesity and T2DM: (1) to look for epigenetic biomarkers that can predict future health problems or pinpoint those who are most at risk, (2) to identify environmental factors associated with obesity that may control gene expression via affecting epigenetic processes, and (3) to aid in the development of new treatment approaches based on dietary or pharmaceutical substances that can change epigenetic pathways [54]. The important tasks at this level are the following: (a) to create robust epigenetic biomarkers of weight regulation; (b) to describe epigenetic markers that are more susceptible to dietary exposures; (c) to recognize the bioactive components that can modify the epigenome; (d) to assess the real importance of certain other obesity-related factors associated with epigenetic regulation; (e) to confirm the stage of life at which the best results are obtained; and (f) to learn about the relationship between obesity and epigenetic regulation [51]. Regarding the aforementioned points, epigenomic profiling of the livers of male offspring whose parents consumed a low-protein diet revealed numerous subtle variations in cytosine methylation based on the paternal diet, which may be connected to variations in the offspring’s lipid and cholesterol metabolism [55].

In the fetal hypothalamus of sheep that experienced mild maternal malnutrition, there was a decrease in the methylation of the proopiomelanocortin (POMC) and glucocorticoid receptor (GR) gene promoters, which may have contributed to long-term energy balance dysregulation. These modifications included altered histone methylation and acetylation, as well as decreased DNA methyltransferase activity [56]. In rodents, being overweight and having excessive fat or sugar absorption are associated with alterations in DNA methylation patterns. These modifications affect the promoter regions of several genes, including LEP [57], NADH dehydrogenase (ubiquinone)1 subcomplex subunit 6 (ND UFB6), and FASN, which have important effects on energy homeostasis and obesity [58]. Numerous studies have shown that reversing the epigenetic markers that are changed by improper food habits and metabolic disorders is a rather simple process. Diet modifies the allele regulation; in particular, TCF7L2 gene expression varies in the pancreatic β cell but is differentially expressed in adipose tissue. The loci of TCF7L2 in exon 4 are involved in the binding of TLE/Groucho corepressors single-nucleotide polymorphism (SNP), in intron 4, and may therefore play a role in the specific expression of isoforms with and without exon 4. Recent studies suggested that whole-grain cereals and foods with a low glycemic index may protect against T2D through the regulation of gene cascade elements in adipose tissue via TCF7L2 allele expression [59].

In human skeletal muscle, Jacobsen et al. [60] showed that a high-fat diet caused alterations in the methylation of 6508 genes, with the highest change in methylation of 13%. After 6–8 weeks on a regular calorie diet, these changes were only partially and insignificantly reversed. Rats’ visceral adipocytes’ CpG 15 was hypomethylated and numerous CpG sites in the LEP promoter region were hypermethylated as a result of a high-fat, high-sucrose diet [61,62]. Changing to a normal caloric diet also reversed the high-fat sucrose-induced DNA methylation alterations at these CpG sites. Additionally, these dietary-related modifications have been linked to hypermethylated CpGs on the promoters of the peroxisome proliferator-activated receptor gamma (PPARG C1A) and FASN, as well as a CpG site of the sterol regulatory element binding transcription factor 1 (SREB F1). These studies provide information about the ability to reverse the phenotypic and epigenetic changes brought on by the consumption of an obesogenic diet. Obesity, insulin resistance, and a higher risk of type 2 diabetes are all linked to the consumption of energy-rich diets. Interestingly, young, healthy males who consumed a high-fat diet for five days experienced changes in their DNA methylation patterns in skeletal muscle [60]. Pathways influencing inflammation, cancer, and proliferation were advanced in 6508 genes with changed DNA methylation levels. The diet was also linked to metabolic changes such as decreased levels of non-esterified fatty acids and hepatic insulin resistance.

## 7. Role of the Gut Microbiota in the Epigenetics of Obesity and Type 2 Diabetes

It has been observed that metabolic diseases are highly associated with alterations in the gut microbial composition. The role of the gut microbiota in epigenetic regulation is shown in Figure 3. The diversity of the microbiota and its derivatives maintain a healthy balance in the intestinal mucosa through DNA methylation, inducing the stimulation of regulatory cytokines and mediators [63]. Environmental and genetic factors play roles in the development of metabolic disorders including changes in the gut microbiota [64]. It is believed that increased penetration of impaired gut membrane by bacterial components induces inflammation through the epigenetic alteration of inflammatory molecules [65]. In addition, microbial metabolites such as Short-Chain Fatty Acids (SCFAs) play significant roles in the regulation of epigenetic programming in various tissues, including the proximal colon, liver, and White Adipose Tissue (WAT) [66].

A study of the microbiota in T2DM subjects revealed a greater abundance of Firmicutes/Bacteroidetes and lactic acid bacteria and a lower concentration of Faecalibacterium prausnitzii in T2DM subjects compared with obese and lean subjects [65]. The study further suggested that changes in the gut microbiota and thus cell wall components are involved in the epigenetic regulation of Toll-like receptors (TLRs) (TLR2 and TLR4). The methylation changes in TLRs were significantly correlated with the body mass index [66]. In obese and T2DM subjects, an analysis of five CpGs in the promoter region of free fatty acid receptor (FFAR)-3 showed significantly lower methylation, and this was highly associated with the least abundant microbiota diversity, mainly regarding Faecalibacterium prausnitzii. The lower methylation of FFAR3 was highly correlated with a higher body mass index [67]. This study also suggested that SCFA-producing bacteria and their products mediate the epigenetic regulation of gene expression. Himanshu Kumar et al. reported, for the first time, that genes with differentially methylated promoters in Firmicutes-dominant microbiota subjects are linked to risk of disease, predominantly cardiovascular diseases and specifically to lipid metabolism, obesity, and the inflammatory response [68]. The differentially methylated regions of diabetes-associated genes UBE2E2 and KCNQ1 in umbilical cord samples of pregnant subjects were significantly associated with the Firmicutes proportion in the maternal gut [69]. The findings of this study show a link between the methylation of diabetes-associated genes in fetuses and maternal microbiota components during pregnancy [69].

Evidence from these studies shows that differences in the gut microbiota composition affect the epigenetic regulation of genes in obesity and type 2 diabetes. An improved diet targeted to induce a gut microbial balance as well as epigenetic changes in pro-inflammatory genes may be effective for the prevention of metabolic syndrome. However, more studies need to further investigate the diagnostic and therapeutic application of the role of the gut microbiota in epigenetic regulation in obesity and T2DM [65].

## 8. Bariatric Surgery and Epigenetics of Patients with Obesity-Associated T2DM

Bariatric surgery is the most effective treatment for obesity. Bariatric surgery is not just an effective method for achieving significant and sustained long-term weight loss; it also intensely improves T2DM, hypertension, dyslipidemia, cardiovascular diseases, and overall mortality. A BMI of 35 kg/m^2^ or higher with one or more ill conditions is the inclusion criterion for bariatric surgery, according to the national institute of health (NIH) [70]. Bariatric surgery provides a chance to investigate the epigenetic mechanism linked with an improved metabolic status related to weight loss [71].

The impact of Bariatric Surgery on the epigenetics of patients with obesity and T2DM has been evaluated by the global profiles of DNA methylation in subcutaneous adipose tissue of Mexican populations [71]. The authors found significant DNA methylation remodeling in adipose tissue six months after the surgery, which was correlated with reductions in body weight and biochemical markers [71]. Whole blood DNA methylation analyses before and after biliopancreatic diversion with duodenal switch (BPD-DS), a metabolic bariatric operation, in obese women showed differentially methylated genes related to diabetic and inflammatory functions (ABCF1, FKBP1A, IFIT2, HTT) [72]. There was a significant correlation between the methylation levels of genes related to insulin action with fasting insulin levels and the homeostatic model of insulin resistance. The authors also reported that differential methylation levels in obese subjects versus treated women provide evidence of durable metabolic improvements after BPD-DS [72]. Recent findings demonstrated that the majority of the genes involved in T2DM and insulin resistance pathways exhibit significant differences in methylation levels, and a majority of these loci were shown to be significantly hypo-methylated in whole blood after BPD-DS compared with a pre-surgery control group [73]. The outcome of whole-genome methylation at CpG sites in obese women after BPD-DS showed an effect of bariatric surgery on the epigenetic signature of genes encoding proteins involved in glucose homeostasis [73]. Glucosekinase-mediated mature onset in young patients is characterized by mildly elevated baseline blood-glucose concentrations that commonly do not require treatment and do not lead to diabetic complications, whereas genetic and polymorphic variants of GCK categories are sensitive to BMI, HbA1C levels, and insulogenic index in patients. These variants are hard to identify and treat [74].

The role of obesity and weight loss after bariatric Roux-en Y gastric bypass (RYGB) surgery was evaluated on peroxisome proliferator-activated receptor γ coactivator-1 α (PGC-1α) and pyruvate dehydrogenase kinase, where isoenzyme 4 (PDK4) is involved in mitochondrial function and fuel utilization in the skeletal muscle of obese women [75]. Romain Barres et al. [75] demonstrated that obesity is associated with hypermethylation of CpG shores and exonic regions close to transcription start sites using a genome-wide DNA methylation analysis of skeletal muscle. The authors reported that the promoter methylation of PGC-1α and PDK4 changed with obesity and was restored to non-obese levels after RYGB-induced weight loss through a global as well as a promoter-specific DNA methylation analysis [75]. The promoter methylation of peroxisome proliferator-activated receptor gamma coactivator 1-alpha (PPARGC1 A), transcription factor A (TFAM), pyruvate dehydrogenase kinase isozyme-4 (PDK4), interleukin-6 (IL6), interleukin-1 beta (IL1β), and tumor necrosis factor-α (TNF) changed two days after RYGB in the blood of obese non-diabetic patients compared with non-obese patients [29]. The author observed similar DNA methylation changes on the day after cholecystectomy and suggested that these changes contribute to improved overall metabolic health after RYGB [29]. Emil K. Nilsson et al. [76] reported that there was not much difference in the mean genome-wide distance between promoter-specific DNA methylation (51 promoters) in whole blood of obese patients at six months after RYGB surgery and controls. These findings of the study supported the idea that RYGB induces genome-wide promoter-specific methylation to improve the molecular effects on metabolic health [76].

According to new findings, obese and lean men have different spermatozoa epigenomes. It is important to note the occurrence of DNA methylation remodeling after weight loss bariatric surgery [77]. A recent study [78] associated surgery-induced weight loss with an intense remodeling of sperm DNA methylation, particularly at genetic locations implicated in the central control of appetite (Figure 4). Men with moderate obesity were shown to have altered epigenetic profiles in spermatozoa and this alteration of the sperm methylome was reversed after bariatric RYGB surgery in morbidly obese humans. These data provide evidence that, under environmental pressure, the epigenome of human spermatozoa animatedly changes and transmits metabolic dysfunction to the next generation [78].

## 9. Links to Other Diseases

Similar to T2DM, changes in epigenetic patterns have been linked to a predisposition to obesity, hypertension, atherosclerosis, and other metabolic disorders. Different non-dietary risk factors are also thought to play roles in epigenetic changes in addition to nutritional ones. Adipogenesis and insulin sensitivity are typically impacted by these variables, particularly hyperglycemia, hypoxia, endocrine disruptors, inflammation, and oxidative stress. It is now well-established that obesity is related to additional diseases, including cardiovascular diseases [79] and many malignancies [80]. There is enticing proof that changes to chromatin may be one method by which obesity confers susceptibility to the advancement of different ailments. For instance, obesity is a noteworthy hazard factor for colorectal cancer, the third most common type of human cancer [81]. The allele variants and histone-mediated allele manipulation inactivate protein tyrosine phosphatase receptor-δ (PTPRD) and subsequent alteration in gliobastoma multiforme (GBM). It is a mutated tumor associated with devastating diseases. These changes showed that PTPRD copy number loss correlates with poor prognosis of PTPRD and basely activates the negative-feedback loop in STAT3 signaling, which is what down-regulates the pathway of obesity and insulin resistance [82]. Sudarsan et al. [83] revealed that obesity induces an enhancer profile that more intently resembles colorectal cancer than normal cells by profiling histone modifications in the colon of mouse models of diet-induced obesity and genetic obesity. Precisely how this happens remains under dynamic debate. The chronic inflammation associated with the obesity state is one potential mediator. Indeed, high serum lipid levels are associated with inflammation and other metabolic complications [84].

## 10. Conclusions

We must take immediate action to stop the dramatic rises in obesity, type 2 diabetes, and associated vascular issues that are endangering the public’s health. It will be essential to gain an understanding of these diseases’ molecular causes. Given that these are complicated diseases with numerous genetic and environmental influences, it will be crucial to use integrative methodologies to fully understand the disease development pathways. Additionally, it has been proposed that poor nutrition during pregnancy or in the early years of life may contribute to an abnormal DNA methylation landscape, which profoundly influences the chances of developing metabolic illnesses in later life. Businesses are able to make use of the quickly evolving high-throughput sequencing technologies for profiling chromatin changes and the associated bioinformatics tools. Furthermore, although there are still a number of obstacles involved, there is a lot of talk about the use of epigenetic therapies, such as those that are already being used for some tumors. As previously stated, significant advancements have been made over the past few years, but considerable work remains.

## Figures and Tables

**Figure 1 medicina-58-01366-f001:**
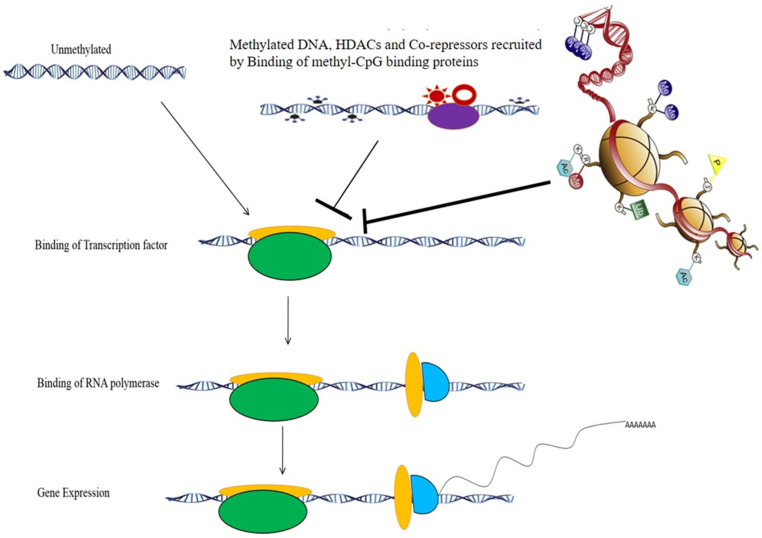
Effect of DNA methylation on gene expression. DNA methylation at the promoter site of a gene recruits the methyl CpG binding protein, which further recruits histone deacetylases and transcriptional repressors, which repress gene expression.

**Figure 2 medicina-58-01366-f002:**
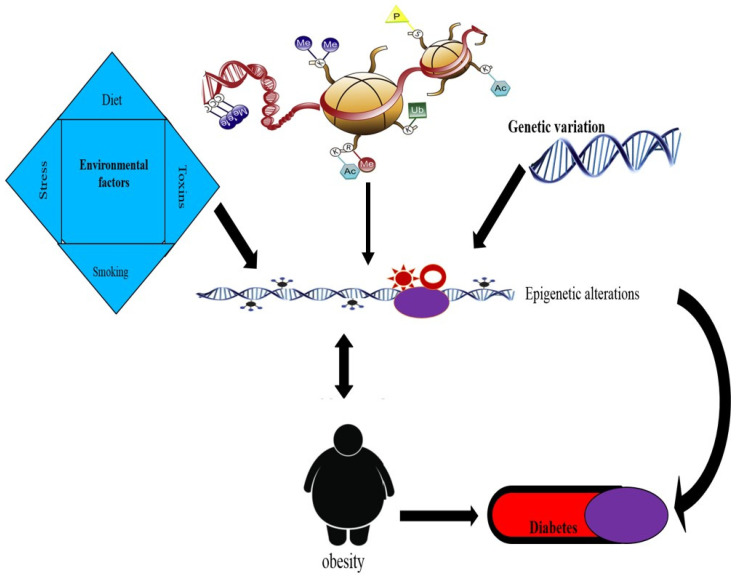
Hypothesis of inter-relationships between environmental factors, genetic variation, epigenetic changes, obesity, and diabetes.

**Figure 3 medicina-58-01366-f003:**
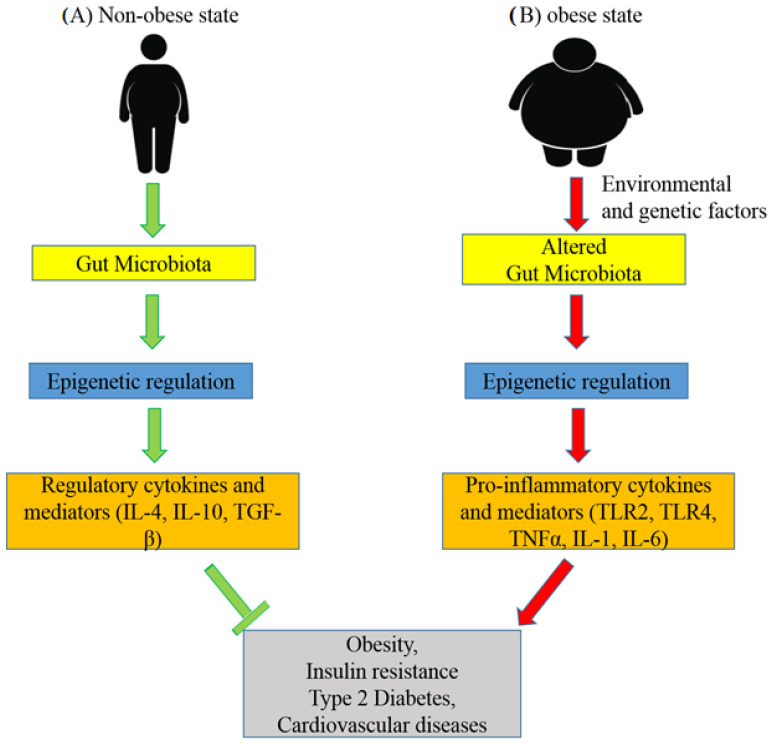
Representative figure of the role of the gut microbiota in epigenetic regulation in non-obese and obese subjects. (**A**) The diversity of the microbiota and its derivatives regulates the stimulation of regulatory cytokines and mediators through epigenetic mechanisms and maintains the healthy balance of the intestinal mucosa. Therefore, the gut microbiota prevents the development of obesity, T2DM, insulin resistance, cardiovascular diseases, and other metabolic diseases. (**B**) Environmental and genetic factors play significant roles in the alteration of the gut microbiota. This increased penetration of impaired gut microbial components induces inflammation through the epigenetic alteration of inflammatory molecules and results in the development of metabolic disorders.

**Figure 4 medicina-58-01366-f004:**
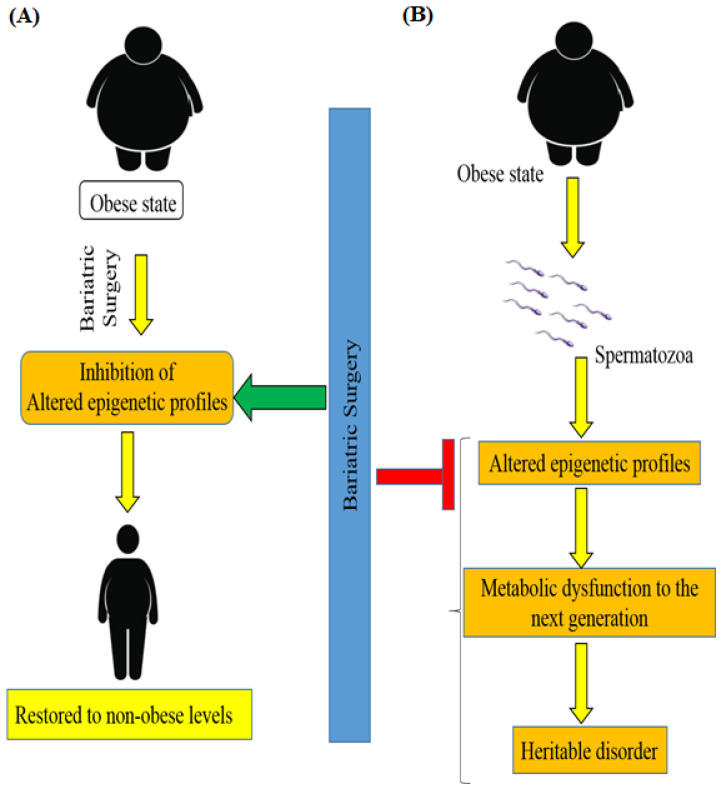
Schematic diagram showing the impact of bariatric Surgery on the epigenetics of patients with obesity and T2DM. (**A**) Bariatric surgery significantly inhibits the alteration of DNA methylation and restores the altered epigenetic profiles to non-obese levels in patients with metabolic disorders. (**B**) Additionally, surgery-induced weight loss inhibits the heritability of the disorder by the next generation by altering the epigenetic profiles.

**Table 1 medicina-58-01366-t001:** Epigenetic modifications of T2DM-associated genes.

Sl. No.	Gene	Epigenetic Change	Significance	Reference
1	Peroxiredoxin-2 (Prdx2) and Scavenger Receptor Class A Member 3 (SCARA3)	DNA methylation	Elevated methylation of Prdx2 and *SCARA3* in T2DM	[23]
2	hepatocyte progenitor kinase-like/germinal center kinase-like kinase (HGK)/Mitogen-activated protein kinase kinase kinase kinase 4 (MAP4K4)	DNA methylation	Upregulated expression of HGK promoter methylation frequencies in T2D patients	[24]
3	Insulin (INS) and pancreatic and duodenal homeobox 1 (PDX-1)	DNA methylation	DNA methylation was increased in the islets of T2DM patients as compared with normal control patients	[25]
4	fatty acid binding protein (FABP3)	DNA methylation	FABP3 (fatty acid binding protein 3) methylation process in peripheral white blood cells associated with plasma total cholesterol, insulin sensitivity, and blood pressure	[26]
5	Peroxisome proliferator-activated receptor gamma coactivator 1-alpha (PGC-1α) –(PPARGC1A)	non-CpG methylation	Rapid epigenetic modulation of PGC-1a, which is involved in the development of T2DM and related metabolic disorders.	[27]
6	Stearoyl-CoA desaturase-1 (Scd1)	DNA methylation	Scd1 factors regulate expression in hepatocytes via altered promoter methylation through a habitual diet.	[28]
7	Pyruvate dehydrogenase lipoamide kinase isozyme 4 (PDK4)	DNA methylation	Roux-en Y gastric bypass increases methylation	[29]
8	PDX1	Histone modification	Chromatin remodeling in the fetus is responsible for the development of T2DM-mediated Intrauterine growth deformities (IUGD)	[30]
9	histone acetylation (HAT) and deacetylation (HDAC) HATs/HDACs or Histone methyltransferases (HMTs)	Histone acetylation	Regulate the gene promoter of islet-specific insulin gene expression in response to changing glucose levels	[31]
10	Jumonji C domain-containing protein (JHDM2A) and Histone H3 lysine 9 dimethylation (H3k9me2)	Histone acetylation	The histone demethylase has been reported to lead to obesity and hyperlipidemia, implying an important role of histone modifications in diabetes	[32]

## Data Availability

Not applicable.

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
