# Peer review of "Epigenetic Regulation of Obesity-Associated Type 2 Diabetes"

_medicina, 2022, doi:10.3390/medicina58101366_

Round 1

Reviewer 1 Report

The review article “Epigenetic regulation of obesity-associated Type 2 diabetes” by Ibrahim et al., provides a comprehensive insight on the epigenetic mechanism of obesity associated T2DM.

The review displays a broad readership to get a summary of epigenetic regulation in T2DM. However, there are several concerns that are summarized below. 

1.     Authors should elaborate the introduction part and include the recent references related to involvement of epigenetic regulations to obesity and T2DM. Several key references are missing including numerous meta-data studies.

2.     I would suggest to elaborate on individual genes presented in Table 1and its epigenetic modifications.

3.     Author should discuss the following genes and shed light upon its modifications from multiple gene studies or meta-data analysis. ARS, PDX-1, GLP-1R, Fosl2, PTPN1, TCF7L2, BCL11A, GCK, MCP-1 MBD2, PTPRD, etc.,

4.     Article should mandatorily undergo language correction by a native English speaker.

Author Response

Authors carefully revised the reviewer comments and Point by point clarifications was attached below.

Reviewer 2 Report

This manuscript reviews the literature that relates the bases of epigenetic regulation with type 2 diabetes and obesity. The review has sufficient quality to be published in Medicina, however, a major review must be done before being accepted. The points to be addressed are listed below:

- The manuscript must be reviewed by a native English speaker, it has many grammatical errors and many sentences are not very clear.

- In my opinion, section 2 should not talk about the relationship of epigenetics with diabetes and obesity, including Table 1. This information should be included in section 4. Even section 3 continues to address general concepts.

- Fig 1 should include histone acetylation as a mechanism of epigenetic modification (this kind of modification must be also mentioned in lines 54/55). In the same figure "co-epressors" must be corrected ("co-repressors"?). In addition, please indicate that the red star and the red ring correspond to transcription repressors.

- Table 1, write the names of the genes following the conventional nomenclature: italics and lowercase.

- Write the full name of the acronyms, and once done, use the acronym in the rest of the text. Homogenize the acronym throughout the text, type 2 diabetes is written as T2D or T2DM. There are many acronyms that do not have the full name on the text, eg: MAPK, Wnt, My018B, etc. (Lines 167-174). Differentiates the nomenclature recommended for proteins (initial capitals and not italics) and genes (italics and lowercase).

- Take care to use a reference in each message that is given within a paragraph. For example: section 3 the reference is only added at the end of the paragraph.

- The font used to cite the figures in the text is not homogeneous, check Fig 1, 2, and 3 in lines 68, 96, and 297, respectively.

- Line 346, write square meters with 2 as a superscript.

- Line 415, double check "ailments".

Author Response

(The authors gave the same response as above.)

Reviewer 3 Report

While this review is covered satisfactorily, the manuscript would be greatly improved by having more clearly defined aims and objectives. The writing is very poor and a native English speaker is required to improve the manuscript. The word count is quite large and there is a need to be more concise. Therefore, please work on these areas and review the following points below.

Abstract: This section is very poorly written. There are major problems with grammatical errors and the structuring of sentences. It is very difficult to follow what you are trying to describe.

Line 31: “WHO has estimated that…”

Lines 32-33: “….90% of T2DM causes were analyzed among 346 million obese subjects with diabetes.” – This is very poorly written.

Line 38: “…which is associated with abdominal obesity,”

Lines 43-44: This sentence could be written more concisely.

Lines 45-47: I cannot understand this sentence. Please revise.

Line 50: “…T2DM is becoming a prominent…”

Lines 68-71: This sentence should be broken up into two.

Figure 1. I think there is a spelling error with “co-epressors”

Table 1: “methylation frequencies”

Line 126: “…positive response mechanisms.”

Lines 134-135: Please revise sentence to improve understanding.

Line 175: “interleukin-7”

Line 177: “…homeostasis model assessment. After….”

Lines 234-235: Abbreviate type 2 diabetes (T2D) as you have done elsewhere.

Figure 3. Please spell out the abbreviations on the figure e.g., TLR, IL-4, TNF etc.

Lines 347-348: “..are inclusion criteria for bariatric surgery according to national institute of health (NIH).”

Line 362-363: “…report demonstrating that…”        

Line 435: “As previously stated, significant…”

Author Response

(The authors gave the same response as above.)

Round 2

Reviewer 1 Report

Author have satisfactorily answered the comments except for the comment 3. Hope it can be done with the minor revision.

Author Response

Reviewer comments

  1. Author have satisfactorily answered the comments except for the comment 3. Hope it can be done with the minor revision.
  2. (Comment-3 - Author should discuss the following genes and shed light upon its modifications from multiple gene studies or meta-data analysis. ARS, PDX-1, GLP-1R, Fosl2, PTPN1, TCF7L2, BCL11A, GCK, MCP-1 MBD2, PTPRD, etc.,)

Author response: Authors carefully revised all the gene and their involvement in diabetes and obesity. The comments was revised various part of the paper from line number- 60-67; 226-234; 254-263; 271-290; 384- 390; 487-492; 542-547. Authors specifically appreciate this comment, the revision of this comment greatly improved the quality of the manuscript.

Reviewer 3 Report

Well done on improving the quality of the manuscript.

Author Response

Authors appreciate the reviewer comments. so thankful to them.

English correction was made again in the whole manuscript.